# Histological and histomorphometric aspects of continual intermittent parathyroid hormone administration on osseointegration in osteoporosis rabbit model

**Yoshifumi Oki**[☯], **Kazuya Doi**[iD]*[☯], **Reiko Kobatake, Yusuke Makihara, Koji Morita, Takayasu Kubo, Kazuhiro Tsuga**

Department of Advanced Prosthodontics, Hiroshima University Graduate School of Biomedical and Health Sciences, Hiroshima, Japan

☯ These authors contributed equally to this work.
* kazuya17@hiroshima-u.ac.jp

**Data Availability Statement:** Data are available from Figshare (DOI: 10.6084/m9.figshare.19519042).

## Abstract

In implant treatment, primary stability and osseointegration are improved by continual intermittent administration of parathyroid hormone (PTH) in patients with osteoporosis. However, the histological and histomorphometric aspects are not clear. The aim of this study was to investigate the histological and histomorphometric effects of intermittent PTH administration on osseointegration in a glucocorticoid-induced osteoporotic rabbit model. Fifteen female New Zealand rabbits were prepared for the osteoporosis model with ovariectomy and glucocorticoid administration. After 1 week, five rabbits were intermittently administered PTH for 8 weeks until the end point (PTHa group) and five for 4 weeks until implant placement (PTHb group). The remaining rabbits were intermittently administered saline for 8 weeks until the end point (Control group). Dental implants were inserted into the femoral epiphyses 11 weeks after ovariectomy. After 4 weeks, the maximum removal torque (RT) of the placed implant and bone implant contact (BIC) ratio were evaluated. In addition micro-computed tomography and histomorphometric analyses were performed. The RT and BIC values were significantly higher in the PTHa group compared with those of the PTHb and Control groups (p< 0.05). Furthermore, the bone mineral densities and Hounsfield units were significantly higher in the PTHa group than those in the PTHb and Control groups. Histologic and histomorphometric measurements revealed that continuous administration of PTH improved bone density and bone formation around the implant placement site, as well as systemic bone formation. Therefore, favorable implant stability was achieved under osteoporosis.

## Introduction

Successful implant treatment depends on the achievement of favorable implant stability. Implant stability comprises primary and secondary stability, including osseointegration [1].

**Funding:** The all authors received no specific funding for this work.

**Competing interests:** The authors have declared that no competing interests exist.

Osseointegration is defined as a direct structural and functional connection between ordered living bone and the surface of a load-carrying implant [2]. Many factors that promote osseointegration have been reported [3], including bone quality and bone quantity which are crucial factors. These factors are influenced by osteoporosis, a disease marked by reduced bone strength. Notably, osteoporosis leads to an increased risk of fractures and broken bones [4]. Bone strength has two primary features: bone density and bone quantity [4]; thus, bone quality and quantity at the placed implant site are factors influenced by osteoporosis. Because bone density at the placed implant site is low in osteoporosis patients, achieving primary stability is difficult [5]. Moreover, it is difficult to achieve osseointegration in osteoporotic sites with low bone quality because of the loss of trabecular bone structure [6].

Glucocorticoids are widely used in the treatment of various diseases, and osteoporosis is a frequent side effect [7,8]. This serous type of osteoporosis is characterized by rapid bone rarefaction with a high risk of fragile bone fractures in the vertebral body and proximal femur [9,10]. Liu et al. observed a significant reduction of the trabecular structure in the femoral condyle of a rabbit with glucocorticoid-induced osteoporosis compared with a healthy rabbit [11]. Indeed, our previous study demonstrated that glucocorticoid-induced osteoporosis reduced the primary stability of implants and mechanical femur bone strength in a rabbit model [12]. Therefore, severe osteoporosis, such as glucocorticoid-induced osteoporosis, may be a risk factor for obtaining primary stability and osseointegration in dental implant therapy. Notably, intermittent administration of parathyroid hormone (PTH) is antagonistic to glucocorticoid-induced osteoporosis because it can enhance bone formation; PTH administration increased the cortical bone thickness and trabecular bone structure in osteoporotic rats [13]. Intermittent administration of PTH is the only clinically authorized therapy that promotes bone formation and is used to treat severe osteoporosis, such as that induced by glucocorticoid administration. Importantly, intermittent administration of PTH is effective for improving low bone mineral density (BMD) at the placed implant site, and for obtaining primary stability and osseointegration in cases of severe osteoporosis, such as those induced by glucocorticoid administration. Our previous study demonstrated that intermittent administration of PTH enhances primary stability in low bone density sites in rabbits with glucocorticoid-induced osteoporosis [14] and evaluated the effects of intermittent continual administration of PTH on bone formation around the implant in a rabbit model of osteoporosis [15]. In that study, however, osseointegration was not evaluated histologically and histomorphometrically; thus, the details are unclear. In addition, there is little information regarding PTH therapy for osseointegration after improved primary stability in low BMD sites, such as those present after glucocorticoid-induced osteoporosis. The aim of this study was to investigate the histological and histomorphometric effects of intermittent PTH administration on osseointegration in a glucocorticoid-induced osteoporotic rabbit model.

## Materials and methods

### Ethics

The experimental plan was approved by the Hiroshima University School (Research Facilities Committee for Laboratory Animal Science, approval no. A-11-5) and was conducted in compliance with the ARRIVE guidelines. All animal experiments were conducted in compliance with the rules of animal experiment in Hiroshima University. Animals were housed in a temperature-, humidity-, and air renewal-controlled room. Animals were fed standard dried diet and water ad libitum. All surgical operations were performed under general and local anesthesia, and all possible efforts were made to minimize suffering during the experimental period.

## Experimental animal procedure

The experimental design is shown in Fig 1. Fifteen 17-week-old (3.0–3.5 kg body weight), female New Zealand White rabbits were used in this study. The animals underwent ovariectomy (n = 15); 2 weeks later, ovariectomized rabbits were injected with methylprednisolone acetate intramuscularly (0.5 mg/kg/day; Depo-Medrol®, Pfizer Inc., New York, NY, USA) for 4 consecutive weeks in preparation for the steroid-induced model of osteoporosis.

Seven weeks after ovariectomy, the animals were classified into three groups. The first group was injected with PTH [1–34] (40 μg/day, 5 days weekly, Forteo®, Pfizer) subcutaneously for 8 weeks (PTHa group, n = 5); the second group was injected with PTH [1–34] subcutaneously for 4 weeks (PTHb group, n = 5). The third group was injected with a saline vehicle solution for 8 weeks (Control group, n = 5). The implant placement surgery was performed 11 weeks after ovariectomy under general anesthesia with medetomidine hydrochloride (0.1 mg/kg, Domitor, Nippon Zenyaku Kogyo Co., Ltd., Fukushima, Japan), sodium pentobarbital (10 mg/kg, Somnopentyl, Kyoritsu Seiyaku Corp., Tokyo, Japan) and local infiltration anesthesia with 2% lidocaine with 1:80,000 noradrenaline (Xylocaine, Dentsply Sirona, Tokyo, Japan). The implant sockets in the distal knee joint epiphysis of both femurs were prepared in accordance with the manufacturer's protocol. After implant socket preparation, the dental implants (diameter: 3.8 mm, length: 6.5 mm; SETiO®, GC Corporation, Tokyo, Japan) were inserted until the color indicator was level with the bone ridge. Four weeks after implantation, the rabbits were euthanized with medetomidine hydrochloride (0.1 mg/kg) and sodium pentobarbital (60 mg/kg). Then they were perfused with 10% neutral formalin through the aorta. The harvested tissue blocks with implants were used for undecalcified histological sections, as previously described [16]. The specimens were immediately fixed in 10% buffered formalin and processed to obtain thin ground sections. Tissue blocks with implants were dehydrated using ascending concentrations of ethanol and embedded in light-polymerized polyester resin (Technovit 7200VLC; Heraeus Kulzer GmbH, Wehrheim, Germany). To achieve complete polymerization of the resin block, photo polymerization equipment was used (BS5000; EXAKT Advanced Technologies GmbH, Norderstedt, Germany). After polymerization, the specimens were sectioned with a high-precision diamond disc to produce a 200-μm-thick cross-section. Undecalcified specimens were ground to approximately 70-μm-thick sections (MG5000; EXAKT Advanced Technologies GmbH), followed by toluidine-blue staining.

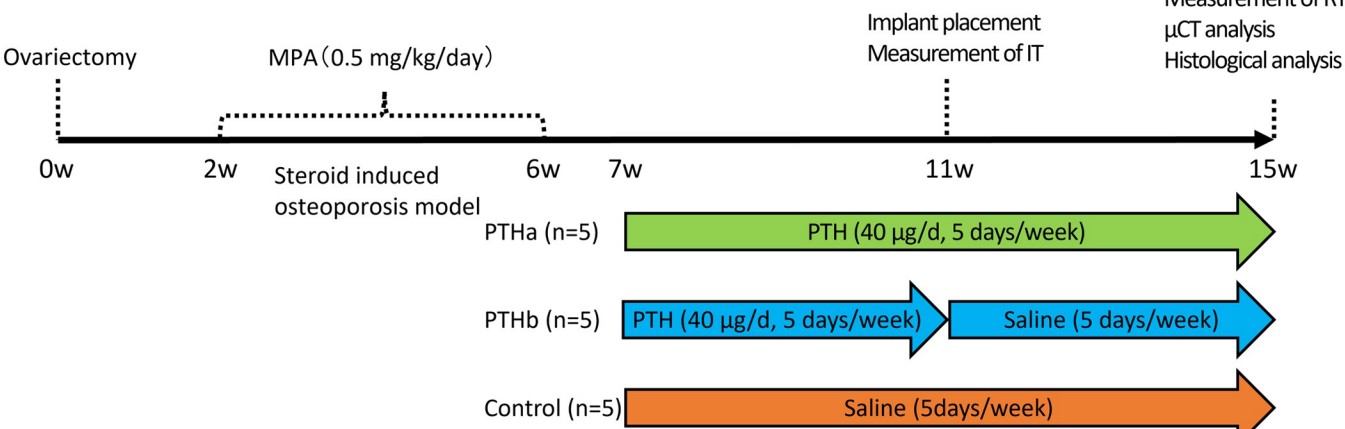

**Fig 1. Study design.** Rabbits underwent ovariectomy (n = 15); 2 weeks later, ovariectomized rabbits steroid injections were initiated for 4 weeks, and the steroid-induced model of osteoporosis was prepared. Seven weeks after ovariectomy, the PTHa group was injected with PTH for 8 weeks; the PTHb group was injected with PTH for 4 weeks; and the Control group was injected with saline for 8 weeks.

Concurrently, the tibiae were harvested, and the tissue blocks were fixed in 10% neutral formalin for 2 weeks. Then, the tissue blocks were cut and decalcified with hydrochloride solution (KC-X®, FALMA, Tokyo, Japan) for 5 days, dehydrated through a graded ethanol series, cleared with xylene, and embedded in paraffin. We obtained 5-μm thick sections from each block and performed hematoxylin and eosin staining.

**Evaluation of primary stability and osseointegration.** During implantation, the maximum insertion torque (IT) was recorded to evaluate primary stability using an implant surgical motor device (iChiropro, Bien-air, Bienne, Bern, Switzerland). Four weeks after implant placement, the maximum removal torque (RT) of femurs with implants (contralateral to those used for preparation of undecalcified histological sections) were measured using a digital torque gauge (BTG-E100CN, Tonichi, Tokyo, Japan), in accordance with a previous study [17]. Furthermore, using the undecalcified histological sections, the bone implant contact (BIC) ratio of each specimen was measured as the ratio of contact length of newly formed bone (total length from the top of the implant shoulder parts to the bottom parts of the first four threads). The BICs were measured using ImageJ (National Institutes of Health, Bethesda, MD, USA).

**Micro-computed tomography.** Before they were used as decalcified sections, the tibiae blocks were scanned on a SkyScan1176 scanner (Bruker, Billerica, MA, USA) and reconstructed with an isotropic voxel size of 9 μm with CTVOX software (Bruker). Images were acquired for the evaluation of cortical bone BMD, as well as the Hounsfield unit (HU) of cancellous bone using computed tomography (CT)-Analysis software (Bruker). The region of interest was from a slice 1.8 mm below the growth plate to the 100th distal slice [18] (Fig 2A).

**Histomorphometric analysis.** A light microscope (BZ-9000; Keyence, Osaka, Japan) was used for the histological analysis of specimens. Histological images of decalcified sections were digitized and analyzed histomorphometrically using ImageJ (National Institutes of Health);

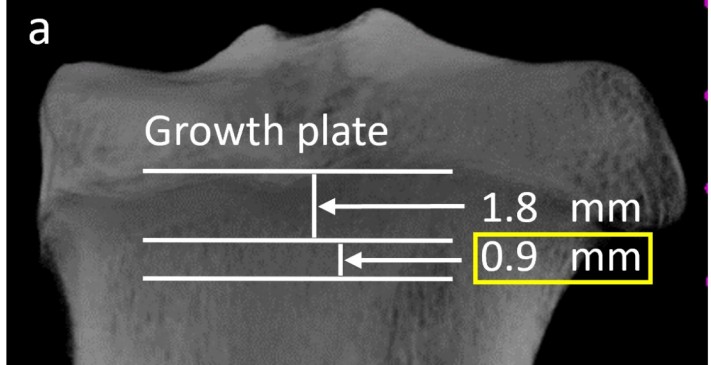
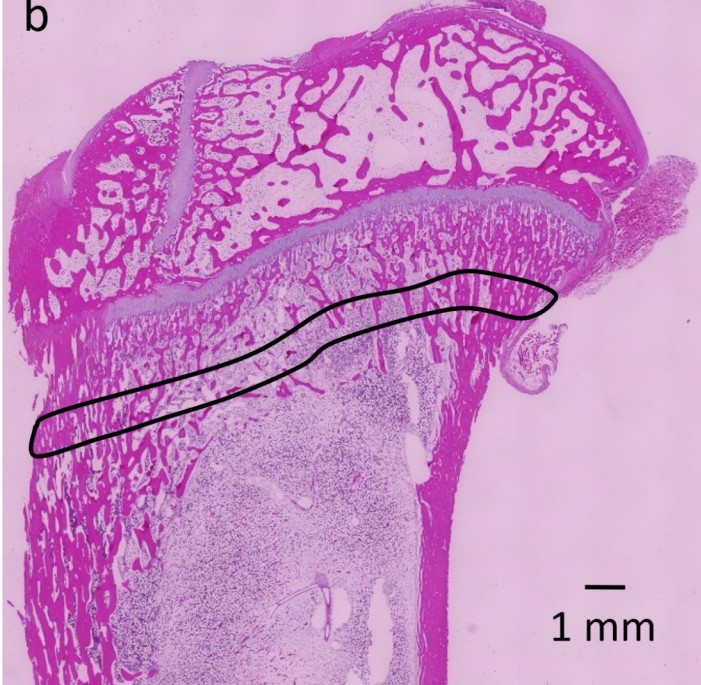

**Fig 2. ROI.** The region of interest (ROI) was from a slice 1.8 mm below the growth plate to the 100th distal slice. (a) ROI of micro-computed tomography analysis and (b) measurement area of histological evaluation.

the bone formation area was measured as the total ratio of cortical to trabecular bone. The regions of interest for the ratio of bone formation calculation were the same as those used the for micro-CT analysis (Fig 2B).

**Statistical analysis.**   All data were presented as means ± standard deviations. One-way analysis of variance and Tukey's post hoc test were performed to assess the presence of any significant differences. P values of <0.05 were considered statistically significant.

## Results

### Evaluation of bone formation at implant placement site

**Histomorphometric analyses.**   The IT results were as follows: 26.9±5.1 Ncm, 27.7±5.3 Ncm, and 8.4±4.2 Ncm in the PTHa, PTHb, and Control groups, respectively (Fig 3). The IT values of the PTHa and PTHb groups were significantly higher than that of the Control group. The RT results were as follows: 77.0±29.2 Ncm, 36.0±8.4 Ncm, and 31.6±15.2 Ncm in the PTHa, PTHb, and Control groups, respectively (Fig 4). The RT values of the PTHa group were significantly higher than those of the PTHb and Control groups. The BIC results were as follows 45.4±17.4%, 39.0±5.7%, and 23.1±9.7% in the PTHa, PTHb, and Control groups, respectively (Fig 5). The BIC values of the PTHa group were significantly higher than those of the PTHb and Control groups.

**Histological evaluation.**   Fig 6 shows the histological undecalcified specimens of the femurs; more new trabecular bone formation was detected around the placed implant in the PTHa group compared with the PTHb and Control groups. A layer of contact without soft tissue between the titanium surface and bone tissue was observed, and good osseointegration was achieved in the PTHa group. In the PTHb and Control groups, bone tissue at the implant

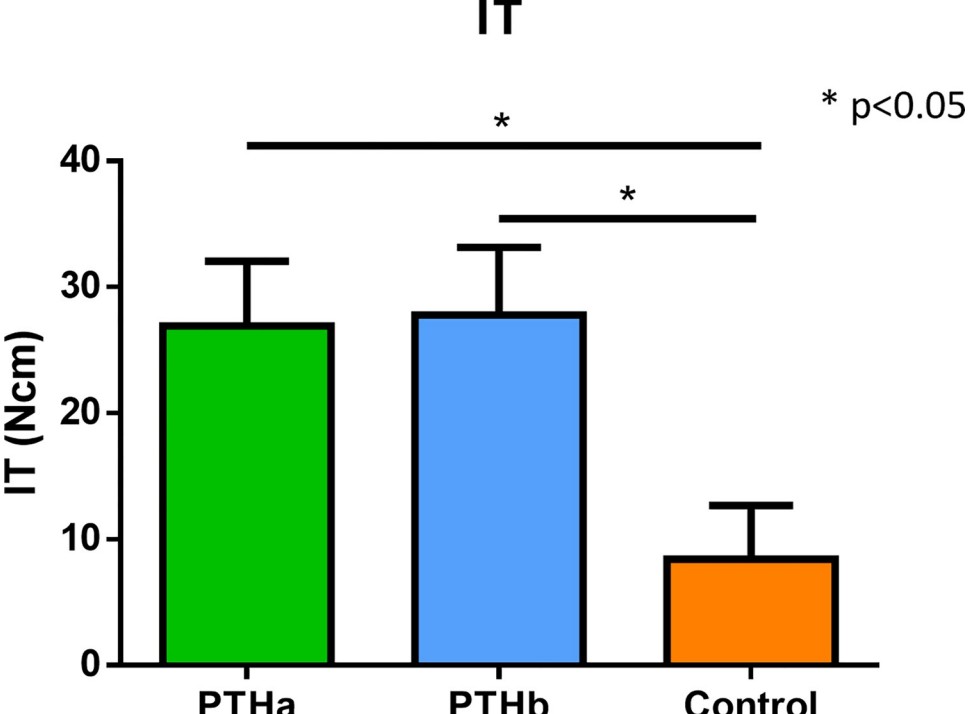

**Fig 3. IT.** The IT values of the PTHa and PTHb groups were significantly higher than that of the Control group. There was no significant difference between the PTHa and PTHb groups.

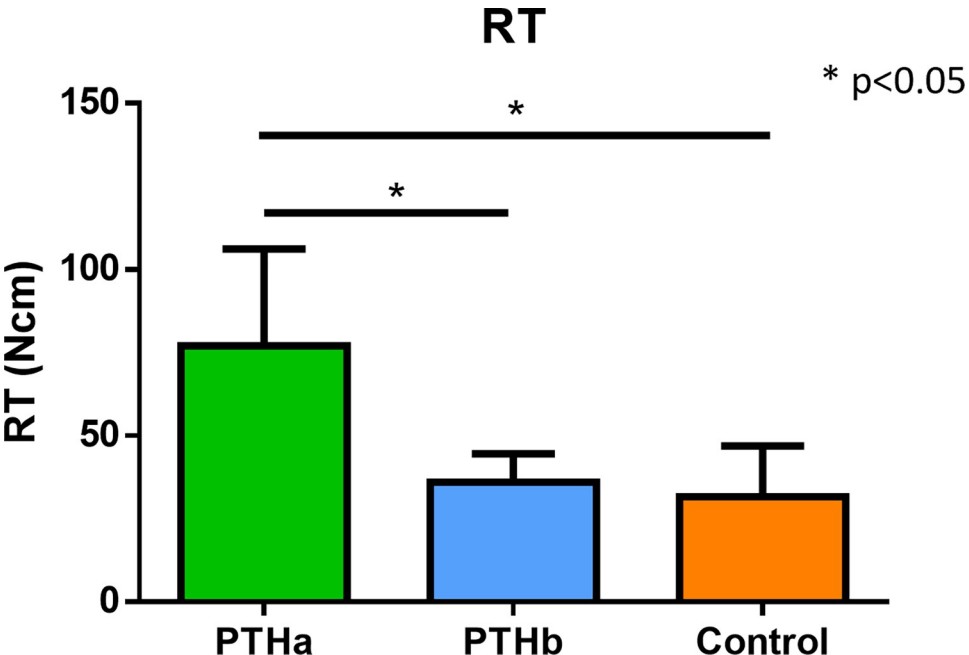

**Fig 4. RT.** The RT values of the PTHa group were significantly higher than those of the PTHb and Control groups. There was no significant difference between the PTHb and Control groups.

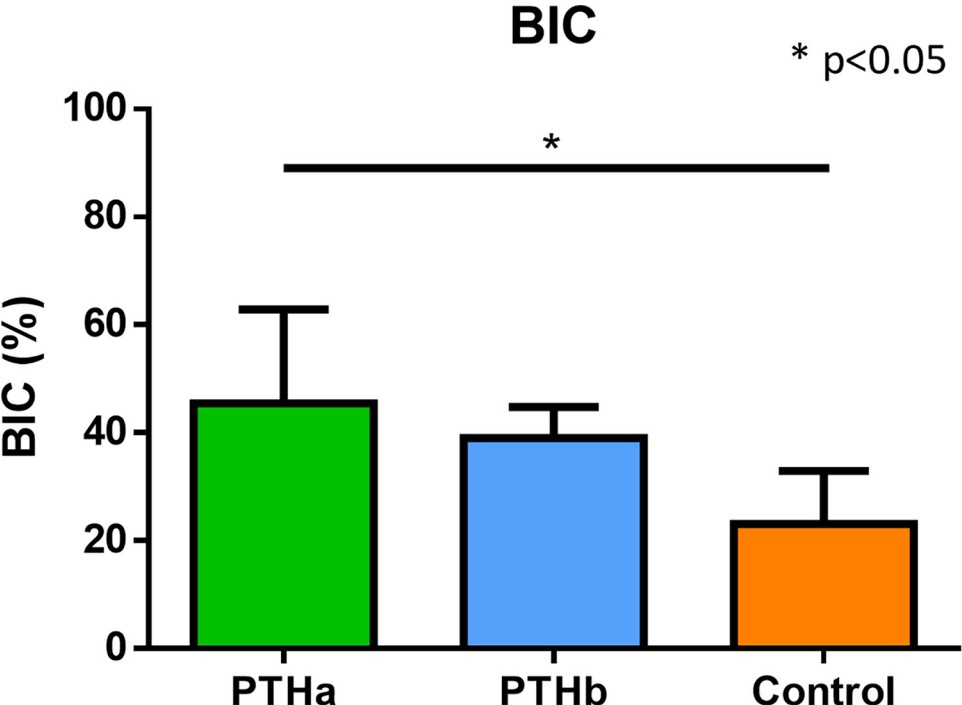

**Fig 5. BIC.** The BIC values of the PTHa group were significantly higher than those of the PTHb and Control groups. There was no significant difference between the PTHb and Control groups.

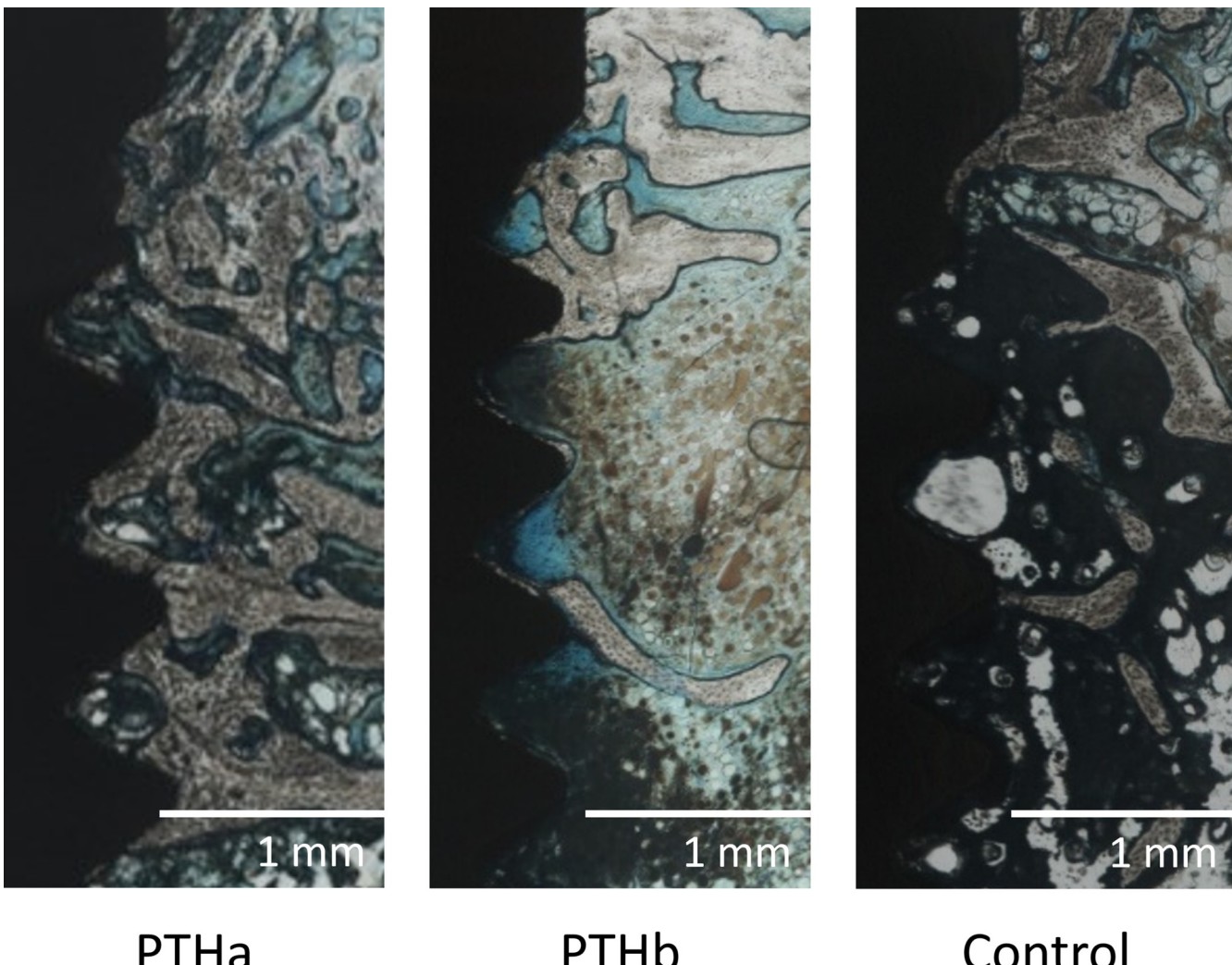

PTHa                    PTHb                    Control

**Fig 6. Histological observation.** In the PTHa group, new trabecular bone formation and bone-implant contact were detected around the implant threads. The PTHb and Control group samples were mostly comprised of connective tissue and bone marrow tissue. Bone-implant contact was partly observed.

surface was observed and osseointegration was partially achieved, although it was mostly comprised of connective tissue and bone marrow tissue.

### Evaluation of bone formation at tibia

**Micro-CT analysis.** Fig 7 shows the micro-CT images of the proximal tibiae. The micro-CT images depicted differences between the PTHa group and PTHb and Control groups in cortical bone and trabecular microstructures. In the PTHa group, the cortical bone width was thick, and the trabecula was seen as a dense structure compared with PTHb group and Control group. The cortical bone BMD results were as follows: 47.8±2.6%, 33.8±18.6%, and 42.2±2.6% in the PTHa, PTHb, and Control groups, respectively (Fig 8A). The BMD value of the PTHa group was significantly higher than those of the PTHb and Control groups. The HU values of cancellous bone were as follows: 1721±37, 1535±68, and 1439±142 in the PTHa, PTHb, and

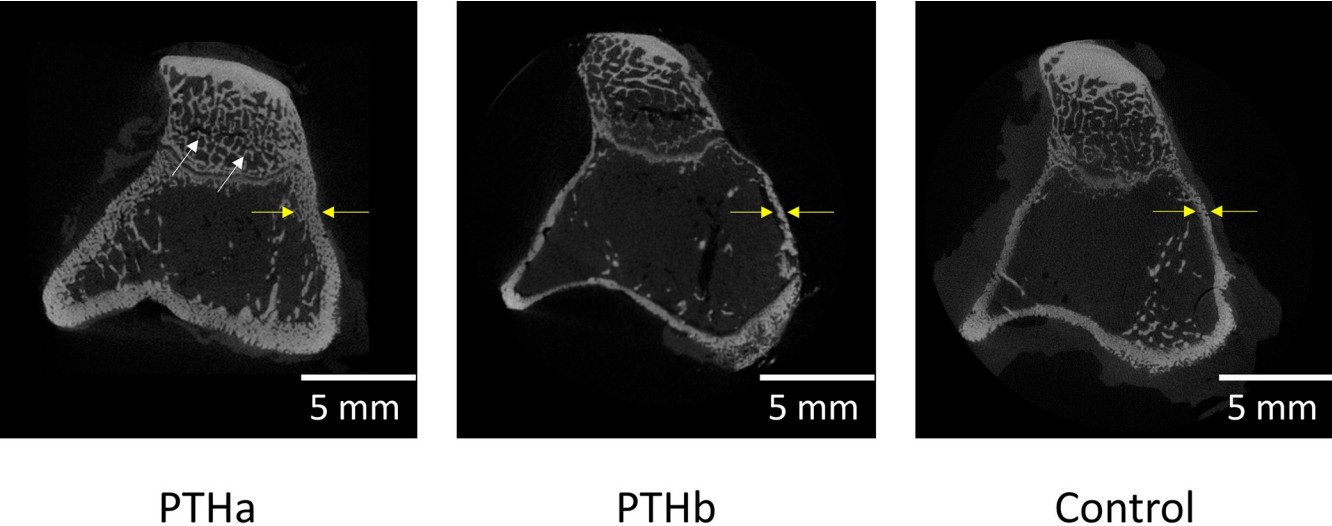

PTHa                        PTHb                        Control

**Fig 7. Micro-CT images of the proximal tibiae.** In the PTHa group, the cortical bone width was thicker than those in the PTHb and Control groups (yellow arrows). The PTHa group had more trabecular bones, which were denser than those in the PTHb and Control groups (white arrows).

Control groups, respectively (Fig 8B). The HU value of the PTHa group was significantly higher than those of the PTHb and Control groups.

**Histological evaluation.**   In the evaluation of the tibiae, many trabecular structures were observed, and they were widely formed in the PTHa group. In contrast, although trabecular structures were observed in the PTHb and Control groups, the amount of formation was small, and the trabecular width tended to be smaller compared with that in the PTHa group.

**Histomorphometric analyses.**   Fig 9 shows the histological decalcified specimens of the tibiae. More trabecular bone structures were found at the epiphysis of tibiae in the PTHa group than in the PTHb and Control groups.

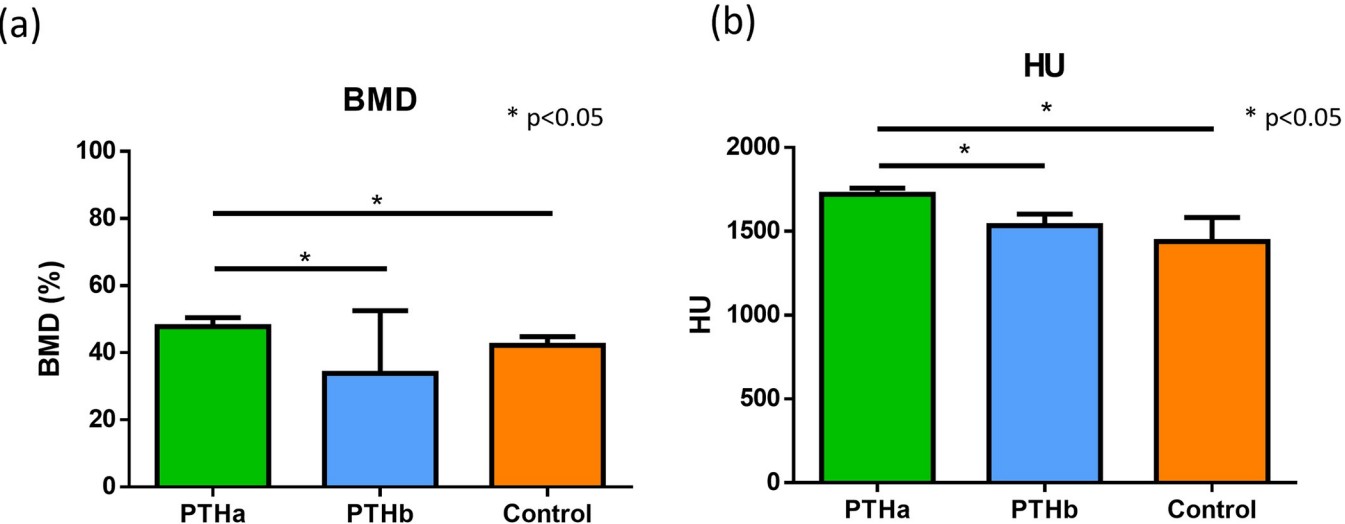

**Fig 8. Micro-CT analysis.** (a) The BMD value of the PTHa group was significantly higher than those of the PTHb and Control groups. There was no significant difference between the PTHb and Control groups. (b) The HU value of cancellous bone of the PTHa group was significantly higher than those of the PTHb and Control groups. There was no significant difference between the PTHb and Control groups.

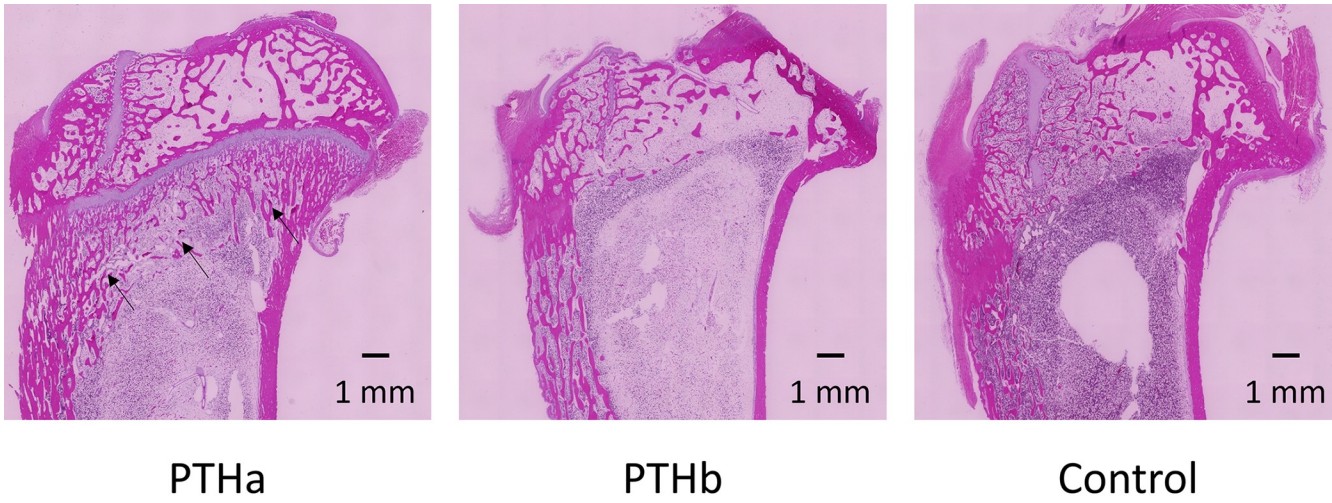

**Fig 9. Histological evaluation of the tibiae.** More trabecular bone structures were found at the epiphysis of tibiae in the PTHa group than in the PTHb and Control groups (black arrows).

The results of the bone formation areas were as follows 29.0±12.3%, 9.1±2.5%, and 14.0 ±7.0% in the PTHa, PTHb, and Control groups, respectively (Fig 10). The ratio of bone formation was significantly higher in the PTHa group than in the PTHb and Control groups.

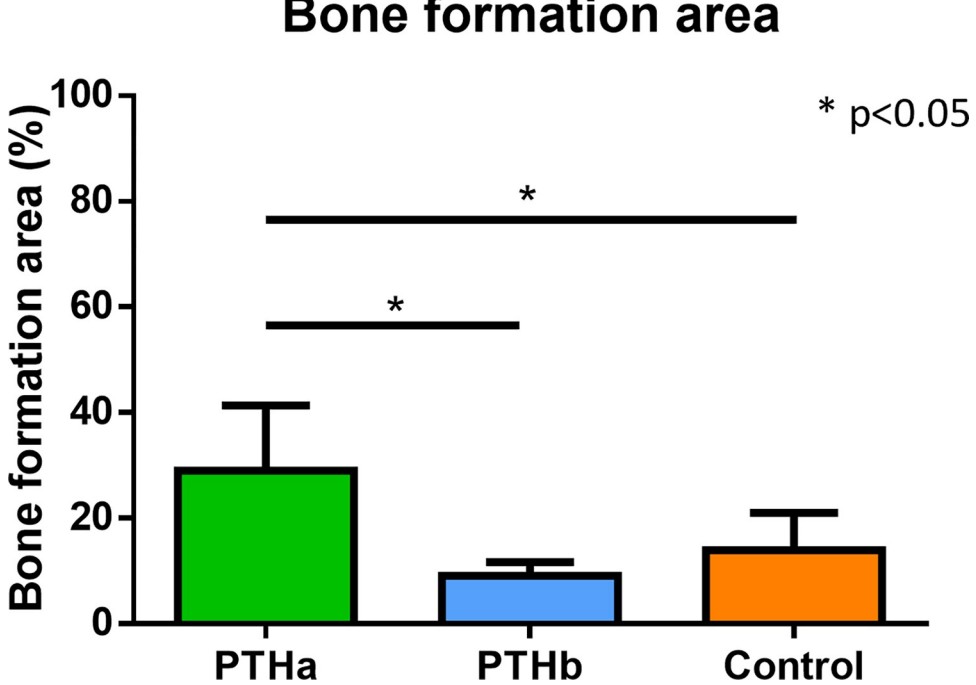

**Fig 10. Bone formation area.** The ratio of bone formation was significantly higher in the PTHa group than in the PTHb and Control groups. There was no significant difference between the PTHb and Control groups.

## Discussion

This study revealed that intermittent administration of PTH can enhance primary stability and osseointegration in a glucocorticoid-induced rabbit model of osteoporosis. Steroid-induced osteoporosis is characterized by sudden drops in bone quality and quantity. Glucocorticoids induce apoptosis of osteoblasts and osteocytes [9], and prolong the lifespan of osteoclasts [9]. Moreover, they control the Wnt signaling pathway that is important in bone formation [19,20] and enhances the differentiation of bone marrow stem cells to fat cells and controls RUNX2, the transcription factor required for the differentiation of bone marrow stem cells to osteoblasts [21]. PTH treatment provides adaptations for severe osteoporosis, such as steroid-induced osteoporosis [22]. Intermittent administration of PTH promotes systemic bone formation by antagonizing the glucocorticoid-induced reduction in RUNX2 mRNA expression [20,23]. In addition, it activates a portion of the Wnt signaling pathway and controls the apoptosis of osteocytes and osteoblasts [24–26]. In our previous study, we revealed that the continuous administration of PTH was effective for achieving primary stability and osseointegration by measuring the implant stability quotient (ISQ) value [15]. The ISQ value was measured using the Ostell® device and is expressed from 1 to 100. A value in the range of 57 to 82 indicates successful implant treatment [27]. Notably, the ISQ value depends on the bone quantity around the placed implant [28,29]; the thickness and strength of the cortical bone and IT increased because the mechanical fit power due to the implant and surrounding bone increased. The osteoporosis model, in which PTH was administered continuously until implantation, showed a good ISQ value at implantation and was effective in achieving primary stability. Furthermore, a comparison between continuous administration of PTH and interruption after implantation showed that continuous administration prevented the decline of ISQ value and was effective for achieving osseointegration. Measuring the ISQ value is a non-invasive evaluation of implant stability; however, it does not directly evaluate the bone condition around the placed implant. Therefore, the IT value is used for evaluation of primary stability, and the BIC and RT values are used for evaluation of osseointegration. The IT value depends on the width and strength of cortical bone [28]. As the IT values were similar to those of our previous study [14,15], the estrogen deficiency due to ovariectomy and glucocorticoid administration caused degradation of trabecular structure at the implant placement site; intermittent PTH administration enhanced the trabecular structure.

Achieving osseointegration is crucial in implant treatment. RT values are expressed as the maximum shear strength during implant removal and depend on the width and strength of cortical bone [30,31]. BIC values depend on the bone quantity around placed implants [32–36]. At 4 weeks after implant placement, the RT and BIC values of the PTHa group were significantly greater than that of the Control group, likely because bone formation at the bone-implant interface was enhanced due to continuous administration of PTH after implant placement, and osseointegration was enhanced. However, there was no significant difference in RT and BIC values between the PTHb and Control groups because promotion of bone formation was lost due to discontinuation of PTH administration after implant placement.

The aspect of local bone connection at this implant placement site was similar to that of systemic bone formation. Systemic bone formation was confirmed by evaluating the tibia. The BMD and HU values of bone formation areas at the tibiae in the PTHa group were significantly higher than those in the PTHb and Control groups. On the other hand, there were no significant differences in BMD and HU values between the PTHb and Control groups. It has been considered that bone remodeling was enhanced due to continuous administration of PTH after implant placement. Notably, the time of PTH action is short, whereas the effect of glucocorticoid administration continues for 3 months after discontinuation [37]. Therefore,

the primary stability after interruption of PTH administration was good in the PTHb group, although the bone formation around the implant was controlled and osseointegration was poor. These results suggest that there is a strong correlation between systemic bone formation and bone formation at the implant placement site.

PTH is used in the treatment of severe osteoporosis, such as steroid-induced osteoporosis, and has the strongest suppressant effect on vertebral body bone fracture among all osteoporotic therapeutic drugs [22]. The administration of PTH comprised intermittent dosage by subcutaneous injection of 40 μg, once per day for 4 weeks, five times weekly, in accordance with previous studies [14,15]. There have been many reports regarding the optimal doses and dosage methods for PTH [38–45]. The common dosage method in previous reports has been intermittent dosage of 15–60 μg per 1 kg of body weight, once per day, five times weekly. Almagro et al. inserted dental implants in a rabbit model of osteoporosis, then began intermittent doses of PTH after implant placement and examined the bone support of the implant body [38]. Notably, implants inserted after bone remodeling were enhanced by PTH treatment. There have been few reports regarding implant placement in severely osteoporotic bone after bone quality was improved by PTH treatment. In the present study, the systemic effects of intermittent PTH administration promoted systemic bone formation in severe osteoporotic conditions, and the local effect of intermittent PTH administration enhanced bone remodeling at the placed implant site, resulting in good osseointegration.

## Supporting information

**S1 Checklist.**
(PDF)

## Author Contributions

**Conceptualization:** Yoshifumi Oki, Kazuya Doi.

**Data curation:** Reiko Kobatake, Yusuke Makihara.

**Formal analysis:** Yoshifumi Oki, Kazuya Doi.

**Funding acquisition:** Yoshifumi Oki, Kazuya Doi.

**Investigation:** Yoshifumi Oki, Kazuya Doi.

**Methodology:** Yoshifumi Oki, Kazuya Doi.

**Project administration:** Yoshifumi Oki, Kazuya Doi, Reiko Kobatake.

**Resources:** Kazuya Doi, Koji Morita.

**Software:** Koji Morita.

**Supervision:** Yoshifumi Oki, Kazuya Doi, Takayasu Kubo, Kazuhiro Tsuga.

**Validation:** Yoshifumi Oki, Yusuke Makihara.

**Visualization:** Reiko Kobatake.

**Writing – original draft:** Yoshifumi Oki, Kazuya Doi.

**Writing – review & editing:** Kazuya Doi, Takayasu Kubo, Kazuhiro Tsuga.

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
