## [Decision Letter · Decision Letter 0]

10 Mar 2022

PONE-D-22-03364Histological and histomorphometric aspect of continual intermittent administration of parathyroid hormone on osseointegration under osteoporosis in rabbit modelPLOS ONE

Dear Dr. Doi,

Thank you for submitting your manuscript to PLOS ONE. After careful consideration, we feel that it has merit but does not fully meet PLOS ONE’s publication criteria as it currently stands. Therefore, we invite you to submit a revised version of the manuscript that addresses the points raised during the review process. Specifically, the manuscript need to be better organized and grammatic errors corrected. Please ensure that your decision is justified on PLOS ONE’s publication criteria and not, for example, on novelty or perceived impact.

We look forward to receiving your revised manuscript.

Kind regards,

Xing-Ming Shi, Ph.D

Academic Editor

PLOS ONE

Journal Requirements:

3. As part of your revision, please complete and submit a copy of the Full ARRIVE 2.0 Guidelines checklist, a document that aims to improve experimental reporting and reproducibility of animal studies for purposes of post-publication data analysis and reproducibility: https://arriveguidelines.org/sites/arrive/files/Author%20Checklist%20-%20Full.pdf (PDF). Please include your completed checklist as a Supporting Information file. Note that if your paper is accepted for publication, this checklist will be published as part of your article.

Reviewers' comments:

Reviewer's Responses to Questions

**Comments to the Author**

1. Is the manuscript technically sound, and do the data support the conclusions?

Reviewer #1: Partly

Reviewer #2: Yes

2. Has the statistical analysis been performed appropriately and rigorously? 

Reviewer #1: No

Reviewer #2: Yes

3. Have the authors made all data underlying the findings in their manuscript fully available?

Reviewer #1: Yes

Reviewer #2: Yes

4. Is the manuscript presented in an intelligible fashion and written in standard English?

Reviewer #1: No

Reviewer #2: No

5. Review Comments to the Author

Reviewer #1: Dear Authors, congratulations for the article ¨Histological and histomorphometric aspect of continual intermittent administration of parathyroid hormone on osseointegration under osteoporosis in rabbit model¨.

In my opinion, several sections should be improved so that the study can be better understood.In the abstract, for example, in the first paragraph the objectives of the study should be presented more clearly. There are too many words repeated in the abstract text unnecessarily. Furthermore, no concrete results are presented or described.

The entire article must be proofread for grammar.

The Materials and Methods and Results sections should be better organized and, please, insert the images where they are mentioned in the text. Figure legends should all be rewritten so that they can be better understood by readers. Using symbols like micro-CT (µCT) is not suitable in many places where it was used and how it was used.

The discussion chapter is very scattered and confusing, it should be rewritten and better organized.

Finally, the authors should present more up-to-date references, several of the references are more than 10 years old.

Reviewer #2: I had the pleasure of reviewing the submission titled "Histological and histomorphometric aspect of continual intermittent administration of parathyroid hormone on osseointegration under osteoporosis in rabbit model". It is scientifically sound and meets PlosOne publication criteria. The manuscript needs to be reviewed for English grammar and style.

6. PLOS authors have the option to publish the peer review history of their article (what does this mean?). If published, this will include your full peer review and any attached files.

Reviewer #1: No

Reviewer #2: **Yes: **Ricardo B. V. Fontes, MD, PhD

---

## [Author Response · Author response to Decision Letter 0]

22 Apr 2022

PONE-D-22-03364

Histological and histomorphometric aspect of continual intermittent administration of parathyroid hormone on osseointegration under osteoporosis in rabbit model

PLOS ONE

Response to Reviewers

We wish to express our strong appreciation to reviewers’ insightful comments on our manuscript. We feel the comments have helped us significantly improve the present study. We thank for careful reading of our manuscript and constructive criticism. The portions were presented in the revised manuscript with track changes and inserted comments. 

Authors answer 

1. We added the method of anesthesia information in the manuscript with yellow highlights based on https://journals.plos.org/plosone/s/submission-guidelines#loc-animal-research. 

2. We have upload our study’s minimal underlying data on figshare site [10.6084/m9.figshare.19519042].

Review Comments to the Author

Reviewer #1: Dear Authors, congratulations for the article ¨Histological and histomorphometric aspect of continual intermittent administration of parathyroid hormone on osseointegration under osteoporosis in rabbit model¨.

1-1) In my opinion, several sections should be improved so that the study can be better understood. In the abstract, for example, in the first paragraph the objectives of the study should be presented more clearly. There are too many words repeated in the abstract text unnecessarily. Furthermore, no concrete results are presented or described.

Authors answer 1-1)

We revised the abstract chapter with track changes. Also, corrected the graph and included detailed data in the results.

1-2) The entire article must be proofread for grammar.

Authors answer 1-2)

We had proofreading in English and attached file of the Editing Certificate.

1-3) The Materials and Methods and Results sections should be better organized and, please, insert the images where they are mentioned in the text. 

Authors answer 1-3)

We inserted the figure images in the manuscript.

1-4) Figure legends should all be rewritten so that they can be better understood by readers. 

Authors answer 1-4)

We revised all figure legends (Figure1~9).

1-5) Using symbols like micro-CT (µCT) is not suitable in many places where it was used and how it was used.

Authors answer 1-5)

We correct the word (we changed from “µCT” to “micro computed tomography; micro-CT”).

1-6) The discussion chapter is very scattered and confusing, it should be rewritten and better organized.

Authors answer 1-6)

We revised the discussion chapter with track changes.

1-7) Finally, the authors should present more up-to-date references, several of the references are more than 10 years old.

Authors answer 1-7)

We added recent papers.

9. Lane NE. Glucocorticoid-Induced Osteoporosis: New Insights into the Pathophysiology and Treatments. Curr Osteoporos Rep. 2019 Feb;17(1):1-7. doi: 10.1007/s11914-019-00498-x. PMID: 30685820; PMCID: PMC6839409.

10. Compston J. Glucocorticoid-induced osteoporosis: an update. Endocrine. 2018 Jul;61(1):7-16. doi: 10.1007/s12020-018-1588-2. Epub 2018 Apr 24. PMID: 29691807; PMCID: PMC5997116.

Review Comments to the Author

Reviewer #2:

Ricardo B. V. Fontes, MD, PhD

I had the pleasure of reviewing the submission titled "Histological and histomorphometric aspect of continual intermittent administration of parathyroid hormone on osseointegration under osteoporosis in rabbit model". It is scientifically sound and meets PlosOne publication criteria. 

The manuscript needs to be reviewed for English grammar and style.

Authors answer 2)

We had proofreading in English and attached file of the Editing Certificate.

---

## [Decision Letter · Decision Letter 1]

13 May 2022

Histological and histomorphometric aspects of continual intermittent parathyroid hormone administration on osseointegration in osteoporosis rabbit model

PONE-D-22-03364R1

Dear Dr. Doi,

We’re pleased to inform you that your manuscript has been judged scientifically suitable for publication and will be formally accepted for publication once it meets all outstanding technical requirements.

Kind regards,

Xing-Ming Shi, Ph.D

Academic Editor

PLOS ONE

Additional Editor Comments (optional):

Reviewers' comments:

Reviewer's Responses to Questions

**Comments to the Author**

1. If the authors have adequately addressed your comments raised in a previous round of review and you feel that this manuscript is now acceptable for publication, you may indicate that here to bypass the “Comments to the Author” section, enter your conflict of interest statement in the “Confidential to Editor” section, and submit your "Accept" recommendation.

Reviewer #2: All comments have been addressed

2. Is the manuscript technically sound, and do the data support the conclusions?

Reviewer #2: Yes

3. Has the statistical analysis been performed appropriately and rigorously? 

Reviewer #2: Yes

4. Have the authors made all data underlying the findings in their manuscript fully available?

Reviewer #2: Yes

5. Is the manuscript presented in an intelligible fashion and written in standard English?

Reviewer #2: Yes

6. Review Comments to the Author

Reviewer #2: (No Response)

7. PLOS authors have the option to publish the peer review history of their article (what does this mean?). If published, this will include your full peer review and any attached files.

Reviewer #2: **Yes: **Ricardo B. V. Fontes

---

## [Editor Report · Acceptance letter]

27 May 2022

PONE-D-22-03364R1 

Histological and histomorphometric aspects of continual intermittent parathyroid hormone administration on osseointegration in osteoporosis rabbit model 

Dear Dr. Doi:

I'm pleased to inform you that your manuscript has been deemed suitable for publication in PLOS ONE. Congratulations! Your manuscript is now with our production department. 

Kind regards, 

on behalf of

Dr Xing-Ming Shi 

Academic Editor

PLOS ONE